# 3D-Bioprinting Strategies Based on In Situ Bone-Healing Mechanism for Vascularized Bone Tissue Engineering

**DOI:** 10.3390/mi12030287

**Published:** 2021-03-08

**Authors:** Ye Lin Park, Kiwon Park, Jae Min Cha

**Affiliations:** 1Department of Mechatronics Engineering, College of Engineering, Incheon National University, Incheon 22012, Korea; apfjd8520@gmail.com; 23D Stem Cell Bioengineering Laboratory, Research Institute for Engineering and Technology, Incheon National University, Incheon 22012, Korea

**Keywords:** bone tissue engineering, vascularization, mesenchymal stem cell, bone healing mechanism, biomaterial, bioprinting

## Abstract

Over the past decades, a number of bone tissue engineering (BTE) approaches have been developed to address substantial challenges in the management of critical size bone defects. Although the majority of BTE strategies developed in the laboratory have been limited due to lack of clinical relevance in translation, primary prerequisites for the construction of vascularized functional bone grafts have gained confidence owing to the accumulated knowledge of the osteogenic, osteoinductive, and osteoconductive properties of mesenchymal stem cells and bone-relevant biomaterials that reflect bone-healing mechanisms. In this review, we summarize the current knowledge of bone-healing mechanisms focusing on the details that should be embodied in the development of vascularized BTE, and discuss promising strategies based on 3D-bioprinting technologies that efficiently coalesce the abovementioned main features in bone-healing systems, which comprehensively interact during the bone regeneration processes.

## 1. Introduction

### 1.1. Needs for Vascularized Bone Tissue Engineering (BTE)

As the median age in our society increases due to increasing life expectancy along with reduced birthrates, the number of bone-grafting procedures for degenerative pathological conditions, tumor resection, and trauma causing large bone defects also increases, thus placing heavy demands on the development of bone tissue engineering (BTE) to construct artificial bone substitutes [1]. Over the past several decades, a wide range of bone-mimetic ceramic compounds using hydroxyapatite (Ca_10_(PO_4_)_6_(OH)_2_), beta-tricalcium phosphate (Ca_3_(PO_4_)_2_), and calcium phosphate cements in different forms have been developed to support the restoration of large bone defects [2,3]. These traditional BTE approaches provided successful results in clinics as they satisfied the macroscopic structural features and mechanical properties required for bone substitutes [4]. However, a lack of proper osteointegration during a long period was observed in the majority of cases, requiring the repetition of surgical interventions in a lifetime [5]. This is because such primitive BTE strategies mainly focused on inorganic bone minerals, only a single type of the tissue compartments constituting bones, disregarding other important factors in the bone-healing system such as cells and signaling cues (i.e., growth factors) [6].

Bone-healing mechanisms are regulated by several features as follows: (1) Varieties of cells including osteoblasts, osteoclasts, osteocytes, osteoprogenitor/precursor cells (i.e., mesenchymal stem cells (MSCs)), and vasculogenic cells; (2) osteoconductive and osteoinductive signaling factors (of which the former is associated with bone growth progressing in the bone matrix and the latter stimulates the recruitment of MSCs and pre-existing osteogenic cells and their differentiation into functional bone cells), secreted by bone-related cells or exogenously derived from circulating blood; and (3) bone extracellular matrix (ECM), which is an organic and inorganic composite scaffold that retains specifically ordered mechanical and biochemical properties with accumulated signaling factors stated above [2,7]. These primary constituents are essential to support the robust structure and function of bone tissues, and bone graft implantation should meet at least one of these properties [8,9].

Traditionally used synthetic bone void fillers may provide an osteoconductive environment to which surrounding host osteogenic cells can migrate and deposit new bone tissue [6,10]. However, as the skeletal system in our body is well-organized with dense vascular beds, bone healing under such an environment cannot efficiently progress deprived of a proper blood supply through full vascularization in the graft [2]. Therefore, osteoconductive environments should include both mitogenic and angiogenic signaling molecules such as insulin-like growth factor (IGF I, II), fibroblast growth factor (FGF), transforming growth factor beta (TFG-β), and platelet-derived growth factor (PDGF) [2,11,12]. In contrast, natural bone ECM and associated signaling molecules can be provided by decellularized bone matrix (DBM) derived from other human donors as a means of allografts, which exhibit both osteoinductivity and osteoconductivity [6]. However, graft procedures using DBM are limited by low donor availability with concerns of possible immune rejection and disease transmission, as well as inefficiency in osteointegration into the host tissue [6,13,14]. Autografts harvested from the patient’s own tissue can circumvent such issues caused by the allograft procedures. Furthermore, autogenous bone grafts contain functional osteogenic cells and bone ECM packed with a cocktail of osteoconductive and osteoinductive factors and a well-organized vascular network pre-existing in the graft. Nevertheless, considerable limitations still exist in the current gold standard, such as limited defect sizes available for autograft, severe pain, infection, and morbidity at the donor site, possibly caused by additional surgical procedures [6,14].

Recently, diverse strategies have been reported in the field of BTE to achieve synthetic, inorganic, and/or biologically organic combinations that recapitulate the physiological structure of native bone tissues through the technical convergence of multiple biotechnologies. Natural bone physiology displays a highly vascularized network made of convoluted blood vessels that are canalized through Harversian or Volkmann’s systems in cortical bone tissues or penetrated within cancellous bone tissues [14]. In addition, the periosteum, a tissue enveloping the outside of the bone structure, is a thin but highly vascularized membrane where osteoblast precursor cells and MSCs are responsible for forming new bone tissue reside (Figure 1) [15]. Blood vessels in bone tissues are essential to supply oxygen, nutrients, and hormones and also deliver supporting or constituting cells to the bone surface required to maintain bone homeostasis and induce osteogenesis during embryonic bone formation or bone fracture repair [16,17,18]. The lack of attention to the vascular networks within bone implant materials could cause long-lasting problems of necrosis at the center of large grafts, resulting in incomplete and inhomogeneous graft viability, which would lead to premature failure of graft integration after implantation [6,13,14]. Accelerating bone function restoration via full graft integration can be achieved by incorporating functional vasculature from the surrounding vascular beds of host tissues deep inside the bone graft [6,19]. Therefore, vascularized BTE constructs would be the future direction for ideal bone substitutes, which should be constructed with a physiological bone environment both osteoconductive and osteoinductive, as well as rich in functional vascular networks.

### 1.2. Importance of MSCs in Bone-Healing Mechanisms

In the early process of intramembranous bone formation, mesenchymal cells aggregate to form a mesenchymal zone, actively attracting the invasion of blood vessel capillaries from the surrounding vascular beds. This process is regulated by patterning osteoinductive signals, provoking MSCs to differentiate toward osteoblasts that deposit the new bone matrix to form the bone blastema [21]. The vasculature invaded into the bony tissue robustly develops and further recruits more bone-constituting cells such as MSCs, osteogenic, and vasculogenic cells. Multiple individual processes simultaneously occur and are eventually fused to form flat bone tissues (Figure 2A) [17]. The bone-healing mechanism resembles this bone developmental process. Adult bone and its adjoining tissues inherently contain less differentiated cells, that is MSCs, other than osteogenic cells, which are crucial in proper bone healing or osteointegration of the implanted bone graft [18]. At the initial part of the bone-healing response, the osteoinduction process instantaneously commences and dynamically directs complex but highly ordered signaling mechanisms, which induce (1) recruitment of MSCs and osteogenic cells, (2) differentiation of MSCs into bone-forming osteoblasts, and (3) stimulation of MSCs to secrete osteoinductive and osteoconductive factors to further support bone restoration (Figure 2B) [2,22]. 

Approximately half a century ago, MSCs were first reported as multipotent cell types isolated from bone marrow, which can differentiate into various mesenchymal lineages such as adipocytes, chondrocytes, osteoblasts, and myoblasts [24]. Since then, MSCs have generated increasing interest in a wide range of biomedical fields. With a growing interest in the therapeutic potential of MSCs, clinical research regarding MSC cell-based therapies have focused on the trophic activities of MSCs that can organize a regenerative microenvironment containing large amounts of bioactive molecules secreted by them [25]. In particular, MSCs have been reported to continuously differentiate into osteogenic cells and supply primary cell sources for bone fracture repair, while pre-existing osteoblasts appear to support fewer portions of the new bone than those derived from MSCs [26]. In addition, MSCs are also known to be populated in one of the supportive cells for microvasculature that comprises endothelial cells (ECs) [27]. Reports have indicated that MSCs cultured in vitro enhance vascular tube formation when implanted in vivo. As the implanted MSCs share angiogenic signaling crosstalk with ECs while expressing vascular endothelial growth factor (VEGF), FGF-2, angiopoietin 1 (Ang-1), and epidermal growth factors (EGFs), the resulting outgrowth of ECs is combined with the host vascular tissues [23,28]. Taken together, to construct the vascularized bone tissue graft, MSCs would be the ideal cell source while acting as the major resource of osteogenic progenitors that form mineralized bone matrix and simultaneously secrete angiogenic factors to stimulate vascularization and stabilize the connective vascular network.

## 2. Mimicking In Situ Bone-Healing Mechanism

### 2.1. Multiple Growth Factors Required in Bone-Healing Mechanisms

When a bone fracture occurs, there is an immediate immune response as immune cells accumulate at the injury site by upregulating the expression of multiple inflammatory growth factors such as tumor necrosis factor-α (TNF-α), FGF, interleukin-1 (IL-1), and interleukin-6 (IL-6) [23]. Simultaneously, the neo-vasculature is actively generated by angiogenic growth factors such as VEGF, PDGF, BMPs, FGFs, and TGF-βs (Figure 3A) [29]. VEGF is the most studied angiogenic factor and is used in many BTE settings [30]. VEGF stimulates the proliferation and migration of ECs, inducing vascular structure formation [31]. After bone fracture, low oxygen tension in the injury site induces hypoxic conditions by hypoxia inducing factors (HIFs), and VEGF is upregulated to promote angiogenesis, thus being pivotal in bone regeneration [31,32]. Odedra et al. reported that the migration of ECs was effectively stimulated by immobilized gradients of VEGF on porous collagen scaffolds [33]. Furthermore, VEGF is also involved in the signaling pathways for osteogenic differentiation of MSCs [23]. It was reported that VEGF embedded in beta-tricalcium phosphate resulted in increased infiltration of microvasculature and osteointegration in a murine calvarial defect [34]. FGF-2 and VEGF contribute to vascular tissue development and growth, thereby promoting wound healing and tissue repair in vivo [23,35]. Similar to VEGF, FGF-2 is also known as an important growth factor that regulates proliferation of ECs and osteoblasts [36]. In addition to VEGF and FGF, multiple growth factors have been reported to contribute to the proliferation, migration, and differentiation of osteoprogenitor cells and concurrent stimulation of angiogenesis in a coupled manner, including PDGF, TGF-β, IGF, and BMPs [12,35]. In particular, BMPs including BMP-2, BMP-4, BMP-6, and BMP-7 have been frequently used in BTE to promote osteogenic and chondrogenic differentiation of MSCs [37]. Among them, BMP-2 and BMP-7 have been approved by the Food and Drug Administration (FDA) for bone regeneration; therefore, they are currently used in diverse cases of orthopedic treatment [38]. A previous study revealed that a hyaluronic acid-based hydrogel in which MSCs and BMP-2 are jointly added was implanted into a rat calvarial defect model, and resulted in the highest efficacy for bone regeneration compared to the cases in which MSCs or BMP-2 alone were implanted [39]. Furthermore, as BMPs are reported to be crucial in attracting blood vessels during the intramembranous ossification process, they have also been used to promote EC proliferation for neo-vasculature formation [40,41]. 

The interrelation of the aforementioned multiple growth factors has been extensively validated by researchers. They are a complex and diverse cascade of signaling mechanisms that provide multifactorial contributions to the pathways governing proliferation, differentiation, and migration of ECs and osteoblasts needed for bone regeneration (Figure 3A) [32,35]. Several previous studies reported that treatment using a cocktail of growth factors in vivo demonstrated synergistic effects resulting in higher efficiency to regulate osteogenic and/or angiogenic differentiation of MSCs than that of a single growth factor. Duneas et al. reported the synergistic effects of a combination of TGF-β and BMP-7 in a dose-dependent manner to induce endochondral bone formation in vivo [42]. Matsaba et al. also reported that a combined administration of TGF-β and BMP-7 carried by insoluble collagenous bone matrix to a rodent model demonstrated a synergistic, dose-dependent, and temporal upregulation of bone formation-related signaling [43]. In addition, bone recovery in a rat cranial critical size defect was enhanced by co-delivering VEGF and BMP-2, suggesting an interactive mechanism between these growth factors during the early stage of bone regeneration [44]. Owing to the advances in biomaterials and drug-delivery systems, interesting studies have been reported to control the release rate of multiple growth factors during the bone-healing process to maximize efficacy. Shah et al. developed multilayered polyelectrolyte films that could impound potent growth factors for bone regeneration such as BMP-2 and VEGF in physiological amounts and release them in a different ratio over a sustained period [45]. In addition, control over sequential delivery of two osteogenic growth factors, BMP-2 and IGF, was reported to effectively stimulate alkaline phosphatase (ALP) activity and thereby increase mineralized matrix formation in pluripotent stem cells [46].

### 2.2. Elaborate Interplay of Cells in a Complex Signaling Cascade of Bone-Healing Mechanisms

Bone formation and vascularization in bone healing are conducted by the elaborate interplay of biological processes of bone-constituting and supportive cells. Bone formation processes include osteogenic differentiation of MSCs, maturation of matrix, and mineralization [1]. During this process, MSCs proliferate into the areas where neovascularization is highly activated, forming a close physical proximity to blood vessels, and continue to differentiate into osteoblasts contributing to bone matrix synthesis [47]. Previous studies have reported that blood vessel-forming cells such as ECs and endothelial progenitor cells (EPCs) are pivotal in neo-vascularization in the new bone matrix and also in bone cell development and activity during bone healing [47,48,49]. Therefore, proper coordination to arrange the synergistic cellular crosstalk between MSCs and ECs or EPCs is of utmost importance for vascularized BTE approaches. MSCs are versatile in bone regeneration because they proliferate and then differentiate into osteoblasts to act as fundamental components of bone formation, and also support blood vessel formation by stimulating the migration, proliferation, and differentiation of vascular cells while secreting a variety of angiogenic factors such as VEGF, FGF, BMP-2, and IGF [47,49]. Furthermore, they can also differentiate into perivascular cells, thus serving as a main resource of new blood vessels [11,50,51].

Synergistic associations and elaborate interplay among MSCs, vascular cells, and osteoblasts have been reported in several studies (Figure 3B). MSCs secret multiple angiogenic factors including VEGF and increase the survival and growth of ECs, thus affecting rapid vascularization in many types of tissues [52]. ECs have been reported to contribute to triggering osteogenic differentiation of MSCs both in vivo and in vitro, and further affect the upregulated activity of ALP in MSCs [53,54]. Other studies investigated the role of paracrine communication among MSCs, osteoblasts, and ECs through VEGF, such that VEGF initially secreted by MSCs and osteoblasts causes the upregulation of VEGF receptor in ECs, thus stimulating vascularization [55], and enhances BMPs expression in ECs, which in turn affects osteogenic differentiation and bone-forming protein expressions of MSCs [56,57]. Li et al. also demonstrated that a representative potent inducer of osteogenesis such as BMP-2 is secreted by both EPCs and MSCs while they are in conjunction [58]. Such cross-talk between MSCs and ECs was reported to be dependent of the state of MSC differentiation. As osteogenesis of MSCs progresses, the migration of EPCs is reduced due to the lack of EPC chemoattractants such as VEGF and FGF, which are actively secreted by undifferentiated MSCs [59]. In this regard, the induction of vasculogenesis was reported to precede osteogenesis to obtain functional vasculature and bone matrix formation, following bone graft implantation in vivo [60]. In contrast, direct cell-cell contacts via coupling of gap junction proteins trigger more intimate cellular responses between MSCs and ECs [61]. Connexin 43, a gap junction protein expressed by both MSCs and ECs was reported to be important in the functional maintenance of bone tissues (Figure 3C) [62]. Co-culturing MSCs and ECs resulted in promoted angiogenesis through the increased VEGF expression while connexin 43 was highly expressed in both cells [56]. In addition, angiogenic potential of EPCs appeared to be reduced due to the lack of connexin 43, resulting in improper bone remodeling process [63].

### 2.3. Scaffolding Technologies to Mimic Bone-Healing Mechanisms

Bone substitutes aiming to provide clinical successes in orthopedic transplantation surgery should be based on three-dimensional (3D) architectures presenting biological considerations required for bone function restoration. Osteoconductive structural guidance is essential to constructing functionalized bone substitutes and should be employed in combination with osteoinductive factors and multiple cell compounds that are important in bone healing. Scaffolds are 3D frameworks made of biomaterials that can induce and regulate cellular attachment, proliferation, migration, and differentiation (Figure 4) [29,64,65]. The primary task of scaffolds in BTE is to construct a structural and mechanical support for 3D cell–cell interactions, providing a microenvironment that is responsive for osteoinducible cells to attach, function, and produce bone ECM on the surface [66]. Traditionally, various bioceramic materials such as hydroxyapatite, beta-tricalcium phosphate, and calcium phosphate cements have often been used for BTE because they exhibit structural and compositional similarities to mineralized bone tissues [3,67]. In the previous study, Wang et al. fabricated biocompatible nano-hydroxyapatite/polyamide composite scaffolds (n-HA/PA) for BTE and demonstrated extensive osteoconductivity of the n-HA/PA scaffolds with host bone tissues when implanted in rabbit mandibles. In addition, the introduction of MSCs to the n-HA/PA scaffolds resulted in significantly enhanced bone regeneration, especially at the early stage of bone healing [68]. As such ceramic materials show a somewhat brittle nature, lack of degradability, and inefficient processability, other types of biocompatible polymers have been developed as alternatives for BTE. Natural polymers such as collagen, HA, silk fibroin, and gelatin-based materials have shown excellent abilities to complement the structural and biochemical niche in bone healing to promote MSC adhesion, migration, growth, osteogenic differentiation, and new bone matrix formation [69,70,71]. An interesting study reported by Schneider et al. demonstrated that MSCs derived from bone marrow and umbilical cord were stimulated by the 3D-contact with collagen I/III gels and subjected to osteogenic differentiation and matrix mineralization, while showing different patterns of expression and synthesis of ECM proteins from each other. Regardless of the origin, both types of MSCs embedded in the 3D-collagen gels secreted matrix metalloproteinases, migrated into the collagenous matrix, and thus resulted in the matrix contraction and structural intensification, which are needed for bone fracture healing [72]. In contrast, biocompatible synthetic polymers such as poly(α-hydroxy) esters, poly(l-lactic acid) (PLLA), poly(glycolic acid), poly(dl-lactic-co-glycolic acid) (PLGA), and polyurethanes have also been developed to confer higher tunability to mechanical and degradable properties and more efficient processability to mimic physiological bone tissue structures [73,74]. Such natural and synthetic polymeric biomaterials have shown comprehensively beneficial assets to be used in BTE, each one with advantages and disadvantages, such as biocompatibility with minimized toxicity and inflammatory reactions, biodegradability to be substituted with new bone matrix deposition, and mechanical stability to provide structural support when implanted in load-bearing sites [66]. Many previous studies reported the synergetic effects of combinations of natural and synthetic polymers used for scaffolding in BTE. Ren et al. fabricated electrospun nanofibrous meshes using a combination of PLLA and gelatin (1:1 in weight ratio) to construct flexible MSC-sheets. The resulting MSC-sheets demonstrated promoted osteogenic differentiation compared with the control, and significant bone regeneration capability in rat cranial defects [75]. A combination of PLGA and chitosan was also reported to improve osteogenic differentiation and mineralization of MSCs while showing the stronger mechanical property compared to the scaffolds made of PLGA alone [76].

The heterogeneous and anisotropic microstructures of bone tissue require an optimized arrangement of pores in the bone scaffolding. Proper scaffold architecture composed of well-defined porosity, pore size, and interconnection is critical for nutrient and waste transfer, as well as cellular growth and infiltration including vasculature toward the central regions of the scaffold, which would lead to successful osteointegration with the host bone [4,77,78]. Previous studies reported that vascular penetration and ingrowth of newly formed tissue into the scaffold after implantation is largely dependent on its pore architecture, such that pore sizes and interconnections over 600 μm and porosity over 70% could generate efficient neo-vascularization, resulting in enhanced bone regeneration [79,80]. In contrast, pore interconnections smaller than 400 μm constrain vascular penetration [81].

When hypoxia occurs in the bone fracture zone, the bone-healing process can be delayed or hindered. It is known that low levels of oxygen may cause necrosis, apoptosis of neighboring cells, and even bacterial infection. A sufficient supply of oxygen is required for ECM synthesis, cell proliferation differentiation, and migration. Therefore, scaffolds using oxygen-generating biomaterials can relieve cell injury during bone regeneration. Peroxide generates water and oxygen with catalysts in vivo, although slowly and spontaneously [82]. Oxygen-releasing biomaterials comprise biocompatible biomaterials infused with inorganic peroxides or peroxide sodium such as sodium percarbonate, calcium peroxide, and magnesium [83]. Harrison et al. synthesized an oxygen-generating compound, sodium percarbonate, resulting in decreased tissue necrosis and cell apoptosis in murine skin flaps [83]. Recently, 3D modeling of biphasic calcium phosphate scaffolds integrated with calcium peroxide particles showed an increase in ALP activity compared to normal scaffolds with oxygen-poor status, which cause a decrease in ALP activity, inhibiting differentiation of both MSCs and osteoblasts, and bone formation [84]. The byproducts of chemical oxygen production are resistant to bacteria; rather, they can sometimes present oxidative stress to cells [85].

### 2.4. Mechanical Environments to Stimulate the Bone-Healing Process

The emergence of stem cell mechanobiology has introduced a paradigm shift in stem cell biology to study cellular signaling mechanisms responsive to extracellular mechanical cues. One of the studies exemplified that mechanosensitive signaling pathways are triggered according to the measure of the force a cell exerts on its bordering ECMs, which significantly influences its morphogenetic changes [86]. The dynamic nature of the mechanical properties of ECMs in different types of tissues was reported to commit stem cells to a specific lineage differentiation. Previous studies revealed that the fate of MSCs could be regulated by exposing them to a specific matrix stiffness that mimics a certain type of tissue in the absence of inducible growth factors [86,87,88]. In particular, MSCs cultured on polymer gels with a stiffness of approximately 25–40 kPa similar to premature bone matrix, that is, osteoid, displayed multiple evidences to signify the osteogenic fate determination, although only showing the initial guidance toward the developmental lineage [89]. Aiming to be used for vascularized BTE, the control of culture matrix stiffness could be synergistically applied with the regulatory effects of osteoinductive factors, supporting the complete terminal osteogenic differentiation. Likewise, the effects of 3D matrix stiffness on EC behaviors were also studied. ECs embedded in collagen with varying stiffness resulted in differential growth rates of angiogenic sprouts as the matrix stiffness increased from ~175 to ~730 Pa [90]. Sack et al. reported that VEGF activities could be different from the facets of VEGF-EC-matrix tethering modulated by ECM stiffness, providing an insight into matrix stiffness-mediated angiogenesis in tissue regeneration [91].

Blood vessels have been exposed to hemodynamic forces, including shear stress, hydrostatic pressure, and cyclic strain, exerted by blood flow to transport oxygen and nutrients in flexible tube architecture [92]. Shear stress is a tangential force applied to blood vessels, and hydrostatic pressure is the fluid flow pushed out by the blood. Cyclic strain is the circumferential and tensile stress on the vascular wall. In addition to the vessel wall, ECs have always been under the influence of hemodynamic forces. When cells are affected by mechanical stimulation, they can translate the strength to electrochemical signals, described as mechanotransduction [93,94]. Mechanosensitive receptors on the cell surface, such as integrins, ion channels, junction proteins, and cytoskeletal proteins, can convert the mechanical cues they receive to biochemical stimuli. Ando et al. demonstrated that the shear stress induced by blood flow can regulate the Ca^2+^ ion level in vascular ECs [95]. As a result, Ca^2+^ ions released from ECs may activate other proteins of the signaling pathway.

Bones are highly porous, and their hydrate structure, inclusive of vascular canals, and the collagen-apatite matrix are affected by the physical loading of fluid flow. Although the underlying mechanisms are still not fully understood, it has been reported that the osteocytes in the lacunar-canalicular system may have mechanosensitive and mechanotransducible properties for fluid flow in a 3D finite element model [96]. In addition, osteoblastic cells and stem cells are sensitive to physical stimuli such as severe conditions for the repetition of bone resorption and remodeling as a result of body movement or muscular contraction [97]. To study the fate of these cells in mechanical stimuli in vivo, many researchers have developed devices that mimic mechanical deformation in vitro. They have flexible magnitudes, frequencies in cyclic strain, or loading modalities of forces. In particular, for cyclic deformation using a bioreactor with four bending points, osteogenic differentiation of MSCs was promoted in an osteogenic differentiation medium [98]. As reported by researchers who studied bone formation focused on cyclic loading, bone formation was induced by cyclic strain frequency of 0.5~1.0 Hz rather than static or temporal forces, while bone resorption decreased [99,100]. 

### 2.5. Nano/Micro-Scale Engineering Applicable for Vascularized BTE

In our body, natural bone and blood vessels are composed of micro-/nano-scalar components. Their cellular contacts with the neighboring ECM are also at the micro-/nano-scale [101]. Notably, the fate determination of stem cells is overly sensitive to the smallest changes in their microenvironment [102]. Therefore, pre-conditioning of biomaterials at the micro-/nanoscale may have the potential to dictate an advantageous vascularized bone graft approach. Chemically, biomaterials with micro-patterned surfaces are highly hydrophilic due to their upgraded surface energy compared to a smooth surface. The hydrophilic property can improve the entrapment of small proteins or increase cell attachment [103]. 

Nanopatterning approaches have been used in various studies of vascular tissue engineering with the ability to mechanically stimulate the surface ligands of ECs, influencing and directing angiogenesis and vessel formation [104]. Soft lithography is one of the classical micro-/nano- patterning methods of duplicating pre-designed soft molding patterns with a polydimethylsiloxane shaped by pouring a pre-polymer into them [105]. Yim et al. demonstrated that vascular smooth muscle cells were readily influenced by the nanopatterned culture substrate fabricated by soft lithography, as cellular migration was induced with elongated morphology and alignment with a parallel direction of the nano-gratings [106,107]. Intriguingly, vascular smooth muscle cells cultured on the nanopatterned substrate showed a decreased proliferation rate and appeared to mimic the contractile feature of adult blood vessels, which are responsible for vascular physiological functions such as vasoconstriction or vasodilation [107,108]. Photolithography curing the materials by ultraviolet has limitations of pitches below 100 nm. Kim et al. developed a nanopatterned culture matrix inspired by osteon in cortical bone, with uniform nanogrooves with a width of 550 nm [109]. MSCs cultured with this nanopatterned polymer can induce osteogenic differentiation either individually or co-cultured with human umbilical vein endothelial cells (HUVECs) at a ratio of 1:1. Recently, lab-on-a-chip devices, known as microfluidic devices, applying soft lithography have been used to manufacture reproducible vascular bio-microelectromechanical systems at the laboratory scale [110].

Several attempts have been made by BTE researchers to create biomimic scaffolding using microsphere sintering techniques including solvent casting/particle leaching, lyophilization, and gas formation because the bone is an interconnected porous structure, as mentioned above. The solvent casting/particle leaching process is simple and requires no expensive equipment. A mixture of polymer and porogen particles containing uniformly sized and inorganic particles is previously cast, followed by leaching to obtain a porous structure [111]. Only a simple frame can be acquired and the residual solvents left inside materials are damaging for cells because they can denature growth factors [111]. Thadavirul et al. reported that a highly interconnected porous scaffold with a uniform pore size of 378‒435 μm induced proliferation and differentiation of pre-osteoblastic cells in vitro [112]. Lyophilization employs ice crystals from the pre-freezing phase without porogen particles [113]. Baheiraei and his colleagues reported COL/β-TCP scaffolds with large porosity (~95–98%) and appropriate pore size (120–200 µm) using the freeze-drying technique [114]. Bone marrow-derived MSCs cultured in this microporous structure showed successful enhancement of osteogenic differentiation, ECM formation, and vascularization in vivo through ALP activity and H&E staining. The high-pressure gas formation method using CO_2_, a non-toxic and non-flammable gas, does not require the additional rinsing process of organic solvents [111]. When a polymer in a chamber is saturated by high-pressure gas and the pressure is rapidly reduced, pores are arranged inside the polymer owing to the thermodynamic instability. Most scaffolds created by this technique have a diverse pore size in the range of 30–700 µm. Sometimes, microporous architecture with overheating during the gas forming procedure might cause non-interconnected pores in scaffolds [115]. 

Nanostructured scaffolds can achieve biomimetic matrix configurations with the sizes similar to that of collagen fibrils, a major component of native ECM [104]. Phase separation techniques could imitate fibril-like structures with a wide range of sizes from micro to nanoscale, and the self-assembly of major protein components of vascular wall, such as collagen and elastin, could form customized nanopatterns to derive selective attachment and spreading of ECs or vascular smooth muscle cells, supporting neo-vessel formation [116,117]. Nanofibrous scaffolds fabricated by electrospinning can rapidly and economically promote cell adhesion, proliferation, and differentiation compared to other scaffolding nanofabrication such as self-assembly of biocompatible proteins, and phase separation [118]. Casanova et al. developed an electrospun nanofibrous scaffold system, which led to successful osteogenic and angiogenic differentiation of MSCs [119]. Electrospinning technology has been reported to be the most widely used in nano-vascularized BTE, because it allows various fibrous patterns with both high porosity and interconnected pores using electrostatic charge [120]. The nanofibrous biomaterials may vary their properties, particularly fiber diameter and morphology by polymer concentration or molecular weight, solvent viscosity, flow rate, applied voltage density, and even distance between the needle and substrates during electrospinning [121].

## 3. 3D-Bioprinting for Vascularized BTE

Scaffolding based-BTE methodologies have shown innovative concepts that could potentially coalesce the main features needed to construct vascularized bone tissue grafts based on a profound understanding of the bone-healing mechanisms. However, the classical scaffold fabrication methods have led to limited success in realizing the dynamic hierarchical architecture of physiological bone tissues comprising spatially ordered multiple types of cells and vascular networks connected to the host blood vessels passing through transverse channels. Since the 1980s, when 3D-printing technologies were first introduced, manufacturing industries have obtained considerable advantages to address the global demand for the customized fabrication of products with complex geometries and highly ordered architectures [122]. This groundbreaking machinery could increase the processability, flexibility, and adaptability (namely on-demand fabrication) for the radically altering customers’ needs in the beginning of the fourth industrial revolution. Recently, 3D-bioprinting technologies have emerged to provide practical solutions in various medical fields to fabricate biomimetic multicellular tissues with a highly complex microenvironment, such as combining 3D-printing, tissue engineering, developmental biology, and regenerative medicine [123]. Similar to 3D-printing, as an additive manufacturing methodology, 3D-bioprinting can precisely control the complex 3D architectures, spatial distributions, and positioning of multiple compositions in a layer-by-layer manner to deposit biomaterials (called bioinks as they allow printing of living cells), in which cells and signaling cues may be embedded as customizing patient-specific therapies [124].

Medical imaging technologies such as magnetic resonance imaging and computed tomography are imperative tools for acquiring precise information on the erratic configuration of large bone defects at the cell and tissue levels. Once raw imaging data have been obtained, computer-aided design (CAD) and mathematical modeling can be used to process 3D reconstruction rendering followed by thin 2D horizontal slices and digitize the complex tomographic information to the specific measurements of tissue dimensions, which can be directly imported to a 3D-bioprinting module as coordinated by layer-by-layer deposition instructions [125,126].

In addition, suitable printing materials, such as cells and bioinks, as well as physical specifications, such as porosity and mechanical properties, reflecting the biochemical and physical microenvironments of the defective tissues, respectively, need to be selected [127]. This phase is the most important groundwork for the successful 3D-bioprinting of vascularized bone tissue graft, which inherently requires a comprehensive understanding of the bone-healing mechanisms, and complex and heterogeneous bone architecture with the composition of organic and inorganic constituents, as we discussed in the former segments of this article (Figure 5). 

### 3.1. Bioprinting Technologies for Vascularized BTE

Although a variety of 3D-printing technologies have been developed in the manufacturing industry, there are only a few options considered available for the 3D bioprinting process, which should allow the maintenance of biomimetic cell friendliness and preservation of bioactive molecules [126]. The three representative classifications that allow for vascularized BTE approaches are: (1) inkjet bioprinting, (2) laser-assisted bioprinting, and (3) extrusion-based bioprinting approaches (Figure 6). Different technical features of these modalities should meet the central requirements for printing living cells, such as high surface resolution at cell levels, operation under physiologically acceptable temperature, and minimized mechanical interferences upon cell viability possibly caused by shear stress [123,126,128].

Inkjet bioprinting technologies were directly derived from commercially available 2D desktop printers. Recently, diverse types of inkjet bioprinters have been developed to achieve printing of the 3D architecture with an elevator stage for the control of the *z*-axis [126,129]. Inkjet bioprinting is a non-contact printing technique for depositing droplets of cell-laden bioinks onto a biocompatible surface by means of thermally, piezoelectrically, or acoustically actuated nozzles, which provide high printing speed, high resolution, affordability, and wide availability [126]. Inkjet bioprinting allows precise printing of overhanging or tubular structures without sacrificial materials, advantageous to construct complex hollow structures of vasculature [130]. Christensen et al. employed inkjet bioprinting techniques and successfully fabricated Y-shaped tubular constructs using alginate hydrogel while avoiding the collapse of the overhanging part of the tube [131]. However, the printing mechanism of ejecting droplets on the surface requires the use of low-viscosity bioinks, which must be gelated by additional crosslinking steps before the next layer of droplets is deposited to form the pre-defined 3D geometry [129,132]. This would limit the available biomaterials used for bioinks and the number of cells possibly loaded in a droplet. In addition, detrimental effects on cell viability are also a concern, which could be caused by the droplet-ejecting mechanisms based on high thermal energy using heat ranging from 200 to 300 °C or mechanical vibration energy generated by ultrasound or piezoelectric stimulators [126,132]. 

Laser-assisted bioprinting utilizes an energy source of focused laser beam to transfer cell-laden bioinks to the surface of the receiving substrate, which is cell friendly and biocompatible. The laser direct-writing approach provides focused laser energy to the cell-laden bioink, in which the energy-absorbed portions are subsequently deposited on the substrate to form a layer of 3D structure. As the focused laser pulse can be used for geometries below 100 µm owing to the rapid gelation kinetics, high resolutions and thereby high shape fidelity in the bioprinted construct can be achieved [133]. In addition, the nozzle-free (contactless) feature of this bioprinting approach can circumvent the side effects such as nozzle-clogging, limited cell density in bioinks, and post-printing cell viability [129]. However, this platform appears to be problematic in depositing multiple cell types and/or materials, and unsuitable for scaled-up fabrication of large tissues, which is necessary to form the dynamic hierarchical architecture of vascularized bone tissues.

Extrusion-based bioprinting is the most common, affordable, and easy-to-use bioprinting approach among the three classified bioprinting technologies [129]. Bioinks with physiological cell densities can be extruded through microscale nozzles using precisely controlled pneumatic pressure or mechanical compressions within a temperature-controlled system [124,126]. Crosslinking of bioinks can be achieved during dispensing and/or depositing processes by chemical, thermal, and/or photocurable means according to the type of polymer [126]. The nozzles and/or a substrate stage are capable of accurate and fine movement along the 3D orthogonal coordination, namely the *x-*, *y-*, and *z-*axes. Byambaa et al. have printed a mass of multiple fibers consisting of HUVECs, MSCs, and GelMA with a VEGF gradient [134]. The central fiber of the mass construct was composed of GelMA with low methacryloyl substitution, which was degraded at the early time point in culture and formed a perfusable vascular lumen in the construct. HUVECs were aligned and MSCs were differentiated into smooth muscle cells on the inner side of the vascular channel. Another fiber surrounding the perfusable channel was loaded with silicate nanoplatelets to induce osteogenesis of MSCs and the other was chemically conjugated with VEGF to promote vascular spreading of HUVECs. Such locally controlled physical and chemical micro-niches constructed by an extrusion-based bioprinting system could simulate sophisticated bone anatomical structure with vascular channels. In another study, Cui et al. developed a novel approach to engineer a vascularized BTE construct, which included bioprinting of a vascularized bone-mimetic scaffold and subsequent molecular functionalization of the scaffold via multilayered sequential nanocoating of VEGF and BMP-2 to coordinate spatiotemporal angiogenic and osteogenic growth factor delivery, respectively [135]. HUVECs and MSCs were co-cultured with medium perfusion in the vascularized scaffold, and the sequentially embedded growth factors were released by metalloprotease-2 naturally secreted by HUVECs, and thus induced vascular formation of HUVECs and upregulated osteogenic differentiation of MSCs. Extrusion-based bioprinting allows for the use of diverse biomaterials, including natural and synthetic polymers (or even customized by various chemical means) with a very wide range of fluid properties, unlike other bioprinting methods [136]. Different hydrogels with varying viscosities could be separately printed in a single work-piece, and the high viscosity of bioinks can contribute to securing structural support for the printed construct, whereas low-viscosity materials would be more suitable for maintaining cellular viability, migration, and function [126]. The shear stress caused by the dispensing pressure could result in lower post-printing cell viability [137]. However, multiple hydrogels with shear-thinning properties that can protect encapsulated cells from high shear forces are currently available for use in extrusion-based bioprinting, as a result of advanced research on non-Newtonian materials [138]. In addition, the extrusion-based bioprinting platform can incorporate multiple extrusion nozzles and coaxial types of nozzles [129,139]. Luo et al. employed self-made shell/core nozzles to the extrusion-based bioprinting system and successfully printed a hollow tube-structured scaffold using alginate/poly(vinyl alcohol) composite materials, demonstrating the feasibility of coaxial extrusion printing to construct a vascular bed arrangement [140]. This feature can enable the concurrent deposition and patterning of several cell types and biomaterials strictly following the preset CAD design, which should be very practical for accurately shaping the complex architecture of vascularized bone tissues. Although relatively low printing resolution and speed compared to other modalities still remain as technical challenges, extrusion-based bioprinters equipped with multiple nozzles have proven to be a promising approach to fabricate vascularized tissue constructs with multiple cell types [129,141]. 

### 3.2. Bioinks for Vascularized BTE

The selection of appropriate bioinks is also one of the most crucial considerations to realize closely mimicking bone physiology. In general, the design of bioinks for printing cell-based tissue constructs should meet the following fundamental criteria: (1) Biocompatibility that avoids unwanted immune responses in the host; (2) cytocompatibility (or cell-friendly conditions) that maintain cell viability and allows cell attachment, migration, and other innate biological functions; (3) printability (or processability) that is mainly dependent on fluid viscosity, gelation (or crosslinking) behavior, and rheological properties such as shear-thinning characteristic, which are directly related to the ease of handling, high print fidelity, and cell behaviors related to mechanical stimulation [142]. Investigation of the processing frame considering the interplay between bioink viscosity, print velocity, and applied pressure is an essential prerequisite for the successful development of functional and viable 3D tissue constructs [143]; (4) degradability, considering the degradation kinetics and byproduct, is also important for innocuous and complete substitution of the graft with the newly grown host tissue [126,142].

Different sources or synthesizing processes lead to distinct characteristics of bioinks. In general, natural hydrogels encourage extensive bioactivity mimicking a structural and biochemical resemblance to the source of ECMs, and synthetic hydrogels such as pluronic, polycaprolactone, and poly(ethylene glycol) offer high tunability for mechanical properties such as alignment, porosity, tensile strength, and elastic modulus, as well as bioactive cues readily bound by diverse chemical modification methods [126,132,142]. Among the diverse types of natural hydrogels applicable for bioprinting, such as silk, chitosan, collagen, gelatin, fibrin, HA, and decellularized ECM (dECM), gelatin-based hydrogels are the most widely used to fabricate the vascular network in engineered constructs [144]. They naturally have cell-binding motifs to facilitate cell adhesion and migration [145]. Furthermore, because of the large number of backbone side chains available for covalent conjugation with various functional groups, they can be readily functionalized with various chemical moieties to enable photo- and/or chemical-crosslinking, which allows highly efficient tunability for mechanical properties as well as additional bioactivity, as that provided by synthetic polymers [146,147,148]. Gelatin has been used in a variety of biomedical fields in the form of gelatin methacryloyl (GelMA), of which biomechanical properties can be simply regulated by varying the formulation conditions such as gelatin concentration and methacrylation degree [144,149]. In this regard, the use of GelMA to form vascularized bone tissue constructs has been frequently reported in the field of BTE. A study presented an innovative concept to fabricate vascularized bone tissue constructs comprising osteogenic cells encapsulated in the GelMA hydrogel with microchannels lined with ECs [149]. Anada et al. reported a novel 3D-bioprinting strategy to fabricate vascularized bone tissue constructs using the GelMA hydrogel with different properties to mimic the dynamic hierarchical structure of bone marrow and cortical shell of bone tissue [150]. Current advances in the formulation technologies of cell-laden hydrogels have introduced a variety of methodologies for blending two or more bioinks using either natural or synthetic hydrogels, or blending both to develop mechanically tunable hydrogels with tailored bioactivity [129,142]. Jia et al. reported the use of perfused vascular structures fabricated by a 3D-bioprinting technique using bioinks blended with alginate and GelMA available for both ionic- and photo-crosslinking [133]. Colosi et al. also reported the fabrication of functional vasculature tissues lined with human umbilical vein ECs by 3D-bioprinting using a blend of alginate and GelMA [151].

## 4. Summary and Future Perspectives

The increasing incidence of severe bone disorders in this aging society places an exceedingly high demand for the development of functional bone grafts. Various BTE strategies have been developed to address the large bone defects in clinical settings; however, the majority of the recent BTE methodologies are limited by the lack of attention to the perfusable blood vessel networks within the bone graft, which are essential for maintaining homogeneous graft viability and further allowing osteointegration into the host tissue after implantation. In addition, extensive studies conducted in the fields of BTE have contributed to the accumulation of knowledge on the primary requirements for constructing vascularized functional bone tissues. Fundamental constituents in the vascularized BTE are suggested on the basis of a deep understanding of the bone-healing mechanisms: (1) MSCs acting as the major resource of osteogenic progenitors forming mineralized bone matrix and simultaneously secreting angiogenic factors to stimulate the formation of vascular networks; (2) osteoinductive and osteoconductive microenvironments that induce and regulate cellular attachment, proliferation, migration, and differentiation of both MSCs and ECs as providing structural and mechanical support for 3D cell–cell interactions within bone substitutes.

However, traditional BTE approaches focusing on biomaterials and scaffolding are limited to materialize the erratic configuration of defective bone with a dynamic hierarchical architecture comprising multiple types of cells, ECMs, and vascular networks. In this regard, 3D-bioprinting technologies have arisen as a promising strategy in the vascularized BTE, providing a series of beneficial features, as follows: (1) The ability to fabricate complex 3D geometries composed of spatially ordered multiple materials; (2) a comprehensive fabrication platform that can encompass and manage most of the primary prerequisites intimated by bone-healing mechanisms; (3) efficient processability to circumvent onerous recipes with numbers of steps set by experimenters; (4) high reproducibility without inconsistency in every single set of production; (5) automatability and cost-efficiency for large-scale production; and (6) high flexibility and adaptability to customize each fabrication process according to specific needs for patients. 3D-bioprinting technologies utilize bioinks embedding living cells and signaling cues, based on an additive manufacturing methodology that can precisely control a complex 3D architecture, spatial distributions, and positioning of multiple compositions in a layer-by-layer manner. The appropriate selection of bioinks is of utmost importance as it allows the construction of biomimetic microenvironments of bone physiology and define the major factors that determine the degree of printability and print fidelity. For the ideal bioink design for vascularized BTE, optimal blending of natural and synthetic polymers should be considered to reflect the dynamic hierarchical architecture of the particular microenvironment of the large bone defect to be cured. Small amounts of high-aspect-ratio nanoparticles could be incorporated with custom-designed bioinks to further enhance the shear-thinning property of bioinks, the tunability of the mechanical stiffness and bioactivity, and the structural integrity after implantation [138,142]. In addition, the development of dynamic crosslinking strategies could provide differential mechanical properties within a single work piece during and/or after printing [126], which allows versatile usability for constructing the multileveled hierarchical structure of vascularized bone tissue grafts. Multi-nozzle extrusion-based bioprinting has also attracted considerable attention with the ability to simultaneously deposit multiple bioinks on a substrate, which could aid in broadening the design criteria available for 3D-bioprinting of vascularized bone tissue grafts, enabling further precise fabrication with multiple cell types and biochemical cues surrounded by a predesigned branched vascular network. To successfully apply this cutting-edge bioprinting technique for vascularized BTE, several technical issues should be addressed, such as difficulties in efficient integration of different types of hydrogels, slow printing rate, and reduced printing resolution.

After all steps of bioprinting have been completed, there is the tissue-maturation phase to consolidate the overall structural integration of the bioprinted cell-based tissue before transplanted in patients [152]. Traditional static culture methods are typically performed on a monolayer condition without proper mixing of the medium, generating a spatially heterogeneous gradient of physicochemical ingredients in culture, which in turn limits the growth of 3D tissues or organs [153]. Providing a 3D culture environment with efficient mixing while avoiding turbulence generation can enhance the mass transport of oxygen, nutrients, and metabolites in or out of the 3D tissue construct in culture [154,155]. Furthermore, continuous medium perfusion can compromise typical concerns introduced by rapid exhaustion of nutrients and oxygen, and accumulation of metabolic wastes in culture with highly proliferating cells of the 3D tissue construct [156]. Therefore, a perfusion bioreactor system with turbulence-free mixing of culture medium is essential to maturate the 3D-bioprinted tissues as implantable bone tissue grafts over an extended culture period [157,158].

3D-bioprinting technologies have been rapidly evolving as a representative multidisciplinary research field that encompasses developmental biology, tissue engineering, regenerative medicine, and material science. With a deep understanding of bone-healing mechanisms, the optimal designs and processes for 3D-bioprinting can be established to construct the functional, clinically applicable large vascularized BTE grafts. We believe that manufacturing patient-customized bone grafts should become a common clinical standard for orthopedic surgeries in the near future.

## Figures and Tables

**Figure 1 micromachines-12-00287-f001:**
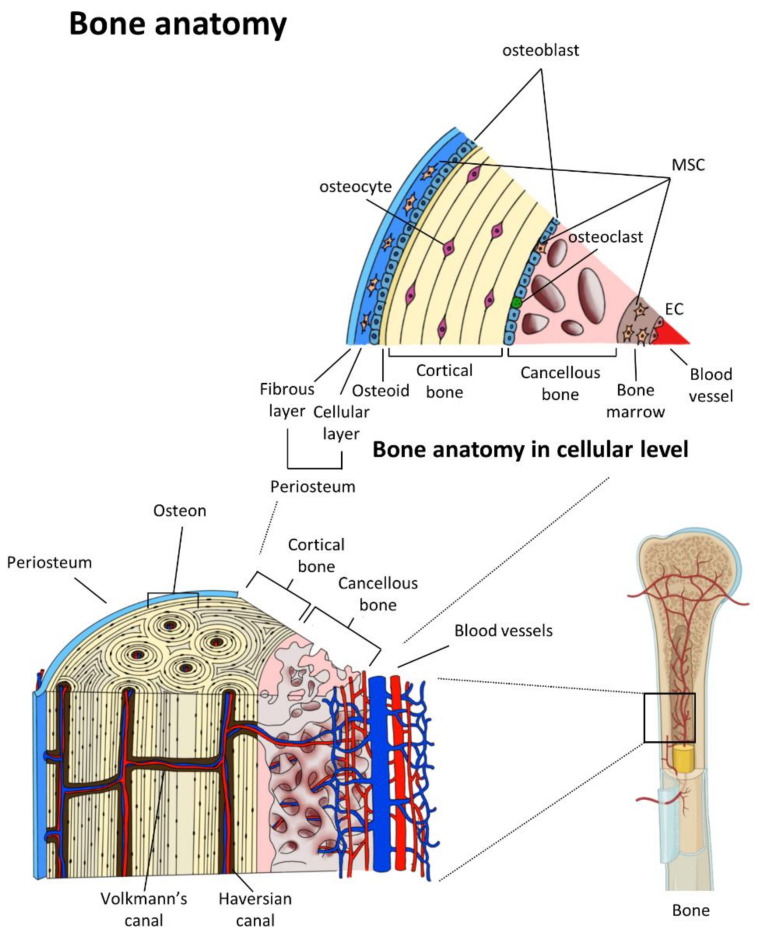
Vascular networks within the hierarchical bone structure and bone anatomy in cellular level. Adapted from [20], published by Springer.

**Figure 2 micromachines-12-00287-f002:**
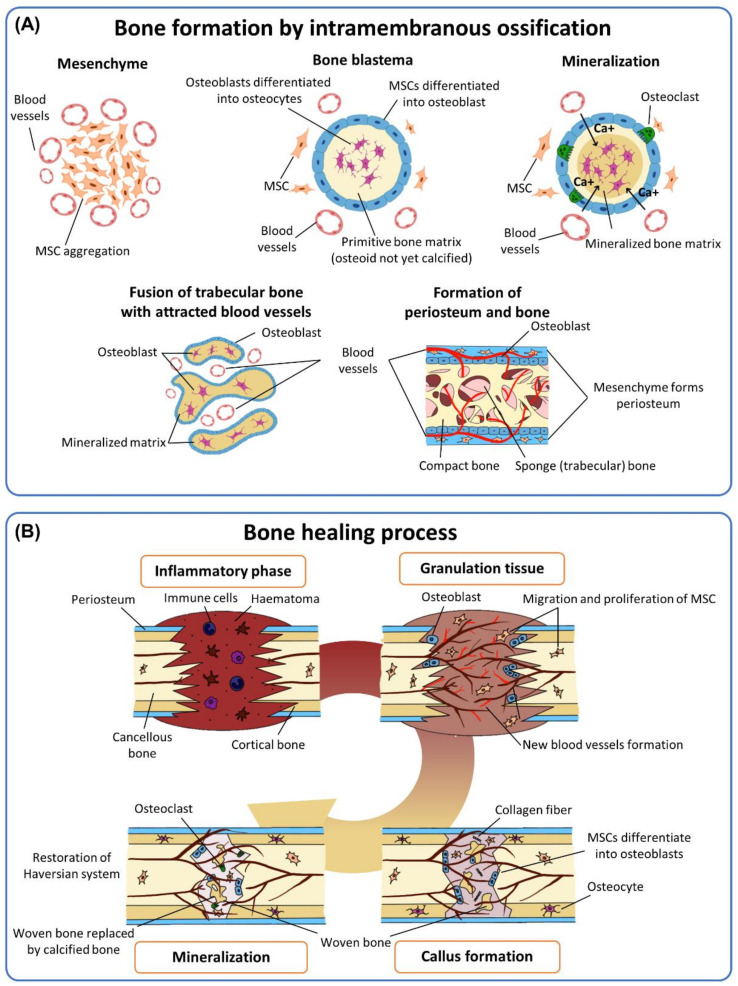
Impact of mesenchymal stem cells (MSCs) in bone formation or regeneration. (**A**) Intramembranous bone formation processes. (**B**) Bone regeneration processes. Reproduced with permission from [23], published by Elsevier.

**Figure 3 micromachines-12-00287-f003:**
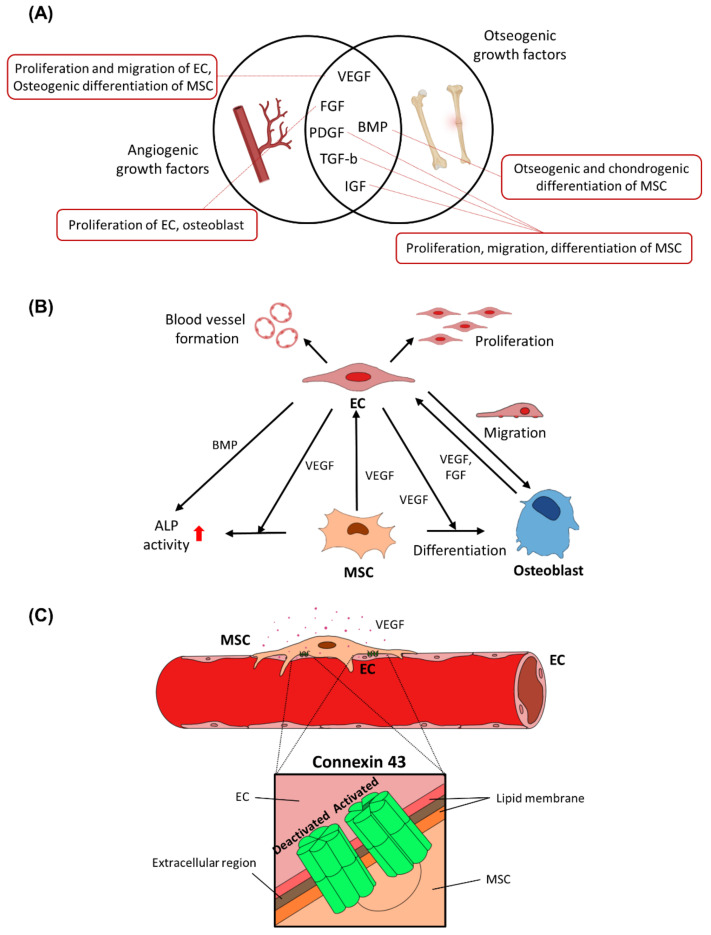
The communication between endothelial cells (EC), MSC, and osteoblast. (**A**) Major growth factors and their functions in bone tissue engineering (BTE). (**B**) The crosstalk interplay between three major cell compounds with growth factors. (**C**) Direct cell-to-cell signaling between MSC and EC via a gap junction protein (connexin 43).

**Figure 4 micromachines-12-00287-f004:**
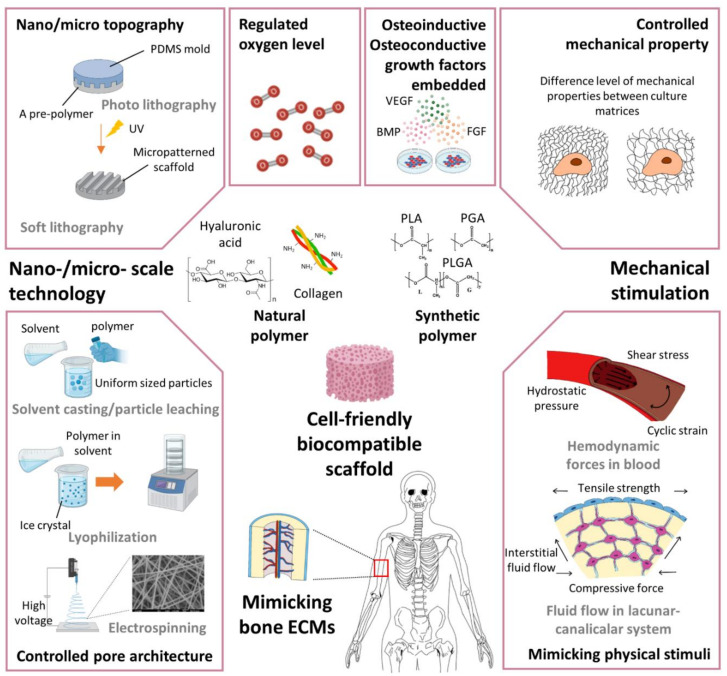
Scaffolding technologies to mimic bone-healing mechanisms using mechanical stimulation, nano-/microscale technology, oxygen tension regulation, and growth factors embedded in scaffolds.

**Figure 5 micromachines-12-00287-f005:**
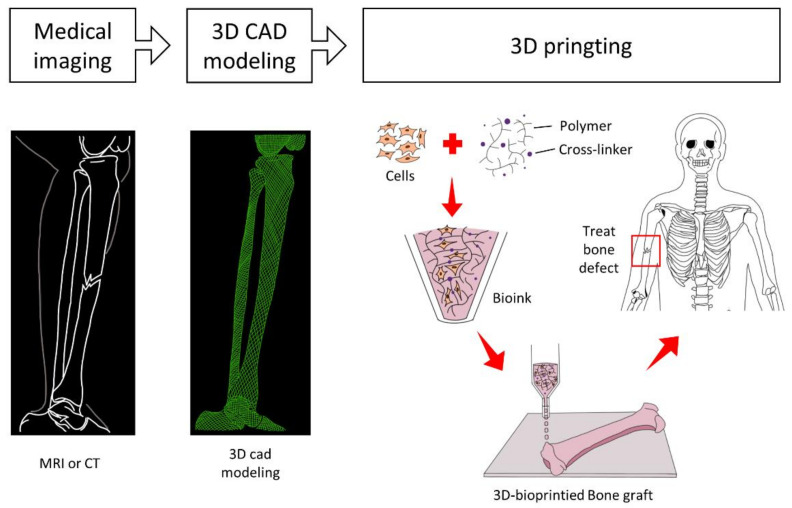
Flow chart of 3D bioprinting of a bone graft. The medical imaging using MRI or CT and 3D-CAD modeling can provide precise dimensional data for a large bone defect, and directly be imported to the 3D-bioprinter. Suitable biomaterials, cells, and printing conditions are determined according to the microenvironments of defected tissues. Finally, a patient can be treated with the elaborate and customized bone graft.

**Figure 6 micromachines-12-00287-f006:**
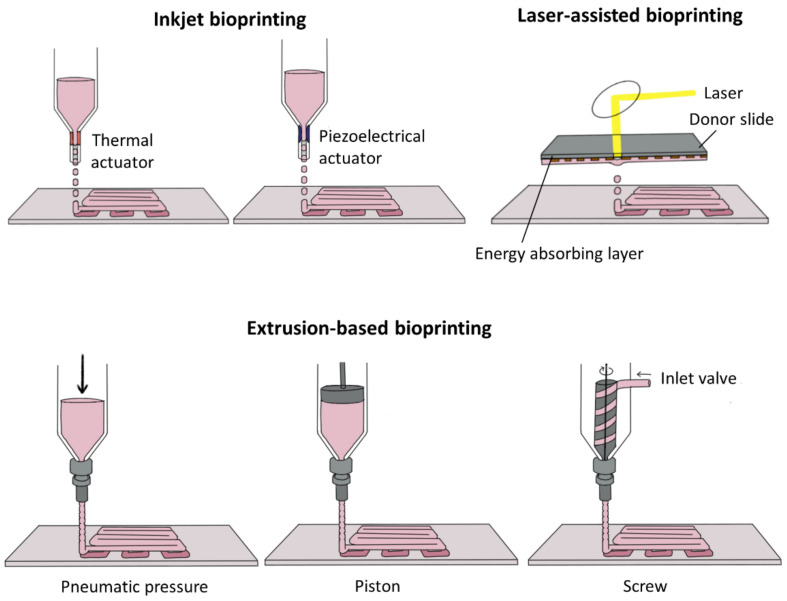
The major 3D-bioprinting methods applicable for vascularized BTE. (1) inkjet bioprining (2) laser-assisted bioprinting (3) extrusion-based bioprinting. Reprined from [129], published by MDPI.

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
