# Peer review of "3D-Bioprinting Strategies Based on In Situ Bone-Healing Mechanism for Vascularized Bone Tissue Engineering"

_micromachines, 2021, doi:10.3390/mi12030287_

Round 1

Reviewer 1 Report

In this review article, the authors discussed the vascularized bone tissue engineering (BTE) facts and covered three main sections. First section is about an introduction to bone tissue engineering and the need for vascularized BTE relatively to the huge demand in bone grafting procedures and bone diseases. In this part the authors introduced the inorganic bone minerals and bone healing mechanisms. Importance of mesenchymal stem cells (MSCs) in vascularized BTE was discussed to cover the differentiation of MSCs into osteoblasts and highlighted the need of vascularized tissue to promote and enhance MSCs differentiation and the role of MSCs in bone healing response. Second part discussed mimicking in-situ bone healing mechanisms and this part covered the growth factors required in BTE such as vascular endothelial growth factor (VEGF) which is the most studied growth factor and fibroblast growth factor (FGF) and their role in stimulating the proliferation and the migration of endothelial cells (ECs). Moreover, discussed the blood vessel forming and its importance in neo-vascularization to form new bone matrix and cell development to enhance bone healing. Scaffolds are the supporting material using for BTE and authors discussed three main points in this article, 1) different materials of scaffolds, 2) mechanical properties of scaffolds and 3) nano/micro scale engineered scaffolds. Authors went through different materials such as natural and synthetic polymers and discussed the effect of the mechanical deformation on MSCs differentiation, and nanopatterning and nanofibrous scaffolds fabricated using electrospinning for promoting cell adhesion. Third section is a review about bioprinting and the different types of bioinks used in vascularized BTE.  I would recommend this manuscript to be accepted after addressing the following comments.

Comments:

  1. Page 1 line 30 after “struct artificial bone substitutes” please add a reference.
  2. Page 1 line 35 after “mechanical properties required for bone substitutes” please add a reference.
  3. Figure 4 page 9 words as GelMA, High voltage are not clear, and please this figure needs to be enlarged in size and to be clear.
  4. Section “2.3. Scaffolding for vascularized BTE” would be important to discuss the differentiation of MSCs with different scaffold materials and the effect and how scaffolds can function in vascularized BTE.
  5. Section “2.5. Nano/micro-scale engineered scaffold for vascularized BTE” needs more discussion regarding the nanoscale vessels. More recent references can be included in this section, please check this reference “Recent advances in 3D bioprinting of vascularized tissues” https://doi.org/10.1016/j.matdes.2020.109398
  6. Overall comment, the paper is specialized in vascular BTE, but it showed a lack of information regarding vascularized tissue engineering, more references should be including in all the sections specially
    • 3D bioprinting for vascularized BTE, this section can add more of the recent advances in vascularized bioprinting and the different types of bioinks used in such experiments, but this section only elaborated the advances in 3D bioprinting and different bioinks in general.

Author Response

Answer notes for reviewers

We sincerely thank the editor and reviewers for their comments regarding our submitted manuscript. The manuscript has been revised and improved by addition, clarification, and correction based on the comments from the reviewers. Responses to the reviewers’ comments along with revisions where appropriate are shown below.

Reviewer 1:

Comment 1. Page 1 line 30 after “construct artificial bone substitutes” please add a reference.

Comment 2. Page 1 line 35 after “mechanical properties required for bone substitutes” please add a reference.

Answer 1 and 2. We appreciate the reviewer’s comments. We have added the appropriate references in the right positions, respectively (please see the revised manuscript highlighted in yellow).

Comment 3. Figure 4 page 9 words as GelMA, High voltage are not clear, and please this figure needs to be enlarged in size and to be clear.

Answer 3. We appreciate the reviewer’s comment. We have improved Figure 4 with clearer words and enlarged illustrations as the reviewer kindly suggested. In addition, other figures have also been revised and improved, accordingly (please see the figures in the revised manuscript).

Comment 4. Section “2.3. Scaffolding for vascularized BTE” would be important to discuss the differentiation of MSCs with different scaffold materials and the effect and how scaffolds can function in vascularized BTE.

Answer 4. We appreciate the reviewer for this valuable comment to improve our manuscript. As the reviewer suggested, we have added correspondent references regarding the effects of scaffold materials on the MSCs’ behaviors in the context of BTE (please see the revised manuscript highlighted in green).

Comment 5. Section “2.5. Nano/micro-scale engineered scaffold for vascularized BTE” needs more discussion regarding the nanoscale vessels. More recent references can be included in this section, please check this reference “Recent advances in 3D bioprinting of vascularized tissues” https://doi.org/10.1016/j.matdes.2020.109398.

Answer 5. We appreciate the reviewer’s comment. In agreement with the reviewer’s point, we have checked the reference the reviewer suggested and added some more contexts and references regarding nanopatterning approaches for vascular tissue engineering (please see the manuscript highlighted in sky blue).

Comment 6. Overall comment, the paper is specialized in vascular BTE, but it showed a lack of information regarding vascularized tissue engineering, more references should be including in all the sections specially - 3D bioprinting for vascularized BTE, this section can add more of the recent advances in vascularized bioprinting and the different types of bioinks used in such experiments, but this section only elaborated the advances in 3D bioprinting and different bioinks in general.

Answer 6. We agree with the reviewer’s comment. We have added more studies and references relevant with bioprinting of the vascular structures (please see the manuscript highlighted in pink).

Reviewer 2 Report

This work is of great interest and the structure of the review is clear but I have two issues with the quality needed for the publication of this work :

First a recent paper deals with the exact same question : 3D Bioprinting for Vascularized Tissue-Engineered Bone Fabrication 10.3390/ma13102278.

The originality of this review is therefore questionable but I think that with all their work, the authors can re write the review with a new highlight in biopriting technologies or the differences between bottom-up  and top down approaches for instance.

Secondly, and it's the major issue: the bibliography is not relevant.

you used two citations for a statement on hydroxyapatite

"Over the past several decades, a wide range of bone-mimetic ceramic compounds using hydroxyapatite (Ca10(PO4)6(OH)2), beta-tricalcium phosphate (Ca3(PO4)2), and calcium phosphate cements in different forms have been developed to support the restoration of large bone defects [1,2]."

The first paper deals with the concept of osteoconducion and the second with cell sheets...it's completely out of the topic of the citation.

a second example

"However, a lack of proper osteointegration during a long period was observed in the majority of cases, requiring the repetition of surgical interventions in a lifetime [3]." This paper is a classic article on the definition of OSSEOINTEGRATION, it's definitely not a paper on the surgical interventions...

"With a growing interest  in the therapeutic potential of MSCs, clinical research regarding MSC cell-based therapies have (has) focused on the trophic activities of MSCs that can organize a regenerative microenvironment containing large amounts of bioactive molecules secreted by them [21]"

this paper commented another article about pericytes.... I cite Caplan : "my suggestion is that all MSCs are pericytes, and this manuscript (Crisan 2018) gives a formal context to better understand, in both embryos and adults, how the MSC/pericyte contributes to the formation, maturation, and homeostasis of all vascularized tissues."

I have a lot of other examples, you used a lot of preview reviews and not the original articles to justify your points. It's a common error with reviews and I'm sure that you can check the bibliography to improve the scientific level of your work.

I truly believe that you must rewrite your review with a new perspective and a relevant bibliography and then you can bring a valuable review for the field of bone tissue engineering.

Author Response

Answer notes for reviewers

We sincerely thank the editor and reviewers for their comments regarding our submitted manuscript. The manuscript has been revised and improved by addition, clarification, and correction based on the comments from the reviewers. Responses to the reviewers’ comments along with revisions where appropriate are shown below.

Reviewer 2:

Comment 1. First a recent paper deals with the exact same question: 3D Bioprinting for Vascularized Tissue-Engineered Bone Fabrication 10.3390/ma13102278. The originality of this review is therefore questionable, but I think that with all their work, the authors can re write the review with a new highlight in biopriting technologies or the differences between bottom-up and top-down approaches for instance.

Answer 1. We appreciate the reviewer for this valuable comment. The paper the reviewer kindly mentioned is a well-written review paper, which we actually have referred to and been cautious not to overlap the contexts as well as structure of our manuscript. The original purpose of our manuscript is to provide the readers with the significant insight into fundamental prerequisites for 3D-bioprinting of vascularized BTE grafts, as accentuating the importance of deep understanding of in situ bone healing mechanisms. Therefore, we have put more efforts on dealing with physiological events that occur when bone fracture is restored and/or regenerated. For example, from the abstract, we tried to explain the purpose of our review paper with the following sentences; “Although the majority of BTE strategies developed in the laboratory have been limited due to lack of clinical relevance in translation, primary prerequisites for the construction of vascularized functional bone grafts have gained confidence owing to the accumulated knowledge of the osteogenic, osteoinductive, and osteoconductive properties of mesenchymal stem cells and bone-relevant biomaterials that reflect bone healing mechanisms. In this review, we summarize the current knowledge of bone healing mechanisms focusing on the details that should be embodied in the development of vascularized BTE, and discuss promising strategies based on 3D-bioprinting technologies that efficiently coalesce the abovementioned main features in bone healing systems, which comprehensively interact during the bone regeneration processes.” And in the introduction section, we have put more efforts to explain the importance of understanding bone healing mechanism to construct vascularized BTE grafts with the following sentences; “Bone healing mechanisms are regulated by several features as follows: 1) varieties of cells including osteoblasts, osteoclasts, osteocytes, osteoprogenitor/precursor cells (i.e., mesenchymal stem cells (MSCs)), and vasculogenic cells, 2) osteoconductive and osteoinductive signaling factors (of which the former is associated with bone growth progressing in the bone matrix and the latter stimulates the recruitment of MSCs and pre-existing osteogenic cells and their differentiation into functional bone cells), secreted by bone-related cells or exogenously derived from circulating blood, and 3) bone extracellular matrix (ECM), which is an organic and inorganic composite scaffold that retains specifically ordered mechanical and biochemical properties with accumulated signaling factors stated above [5,6]. These primary constituents are essential to support the robust structure and function of bone tissues, and bone graft implantation should meet at least one of these properties [3,7].” Further, we also tried to emphasize the importance of MSCs in bone healing mechanisms in the section 1.2, which mainly deals with the contexts that MSCs would be one of the primary constituents for vascularized BTE as they are in the bone healing mechanisms. Beside these efforts, there are many other endeavors we put throughout the manuscript to link the knowledge of bone healing mechanisms and bone physiology/structure to the construction of vascularized BTE grafts (in particular, section 2.1 and 2.2). And, our manuscript ends up with the contexts that such knowledge of bone healing mechanisms is fundamental prerequisites for 3D-bioprinting of vascularized BTE grafts with the following sentences in the summary; “Fundamental constituents in the vascularized BTE are suggested on the basis of a deep understanding of the bone healing mechanisms: 1) MSCs acting as the major resource of osteogenic progenitors forming mineralized bone matrix and simultaneously secreting angiogenic factors to stimulate the formation of vascular networks, 2) osteoinductive and osteoconductive microenvironments that induce and regulate cellular attachment, proliferation, migration, and differentiation of both MSCs and ECs as providing structural and mechanical support for 3D cell-cell interactions within bone substitutes.”

The article the reviewer kindly suggested deals with the contexts of bone physiology and vascularization mechanisms only in a single section “2. Mechanisms of Vascular Formation”, while it appears to more focus on the introduction of various technical approaches. We humbly think that our manuscript is differentiated from the suggested article and also from other review articles in the aspects we exemplified above.

However, as respecting the reviewer’s concerns that should lead to the significant improvement of our manuscript, we have revised and added some expressions and wordings that may have brought about misconception on our manuscript (please see the manuscript written in red). The revised is listed as follows: 1) the title of our manuscript has been changed to “3D-bioprinting strategies based on in situ bone healing mechanism for vascularized bone tissue engineering” to more clarify the purpose of our manuscript; 2) accordingly, the subtitles of 1.2, 2.1, 2.3, 2.4, and 2.5 have been changed; and 3) at the very end of summary section, we have added the following sentence to emphasize the purpose of our manuscript one more time: “With a deep understanding of bone healing mechanisms, the optimal designs and processes for 3D-bioprinting can be established to construct the functional, clinically applicable large vascularized BTE grafts.”

We hope the reviewer could kindly understand our position, and please let us have more valuable advices on any concerns that the reviewer could have in the revised manuscript.

Comment 2. Secondly, and it's the major issue: the bibliography is not relevant.

Answer 2. We appreciate the reviewer for pointing out this important issue. We should have been more careful about this fundamental matter and apologize to cause any inconvenience for the reviewer to review our manuscript. As the reviewer suggested, we carefully reviewed again and revised the bibliography. The following is the table presenting all the revised and corrercted bibliography in the revised manuscript (please see the manuscript highlighted in grey).

Original bibliography

Revised bibliography

As the median age in our society increases due to increasing life expectancy along with reduced birthrates, the number of bone-grafting procedures for degenerative patho-logical conditions, tumor resection, and trauma causing large bone defects also increases, thus placing heavy demands on the development of bone tissue engineering (BTE) to con-struct artificial bone substitutes [1].

[1] none

[1] Amini, A.R.; Laurencin, C.T.; Nukavarapu, S.P. Bone tissue engineering: recent advances and challenges. Crit Rev Biomed Eng 2012, 40, 363-408, doi:10.1615/critrevbiomedeng.v40.i5.10.

Over the past several decades, a wide range of bone-mimetic ceramic compounds using hydroxyapatite (Ca10(PO4)6(OH)2), beta-tricalcium phosphate (Ca3(PO4)2), and calcium phosphate cements in different forms have been developed to support the restoration of large bone defects [2,3].

[2] Miron, R.J.; Zhang, Y.F. Osteoinduction: a review of old concepts with new standards. J Dent Res 2012, 91, 736-744, doi:10.1177/0022034511435260.

[3] Chen, M.; Xu, Y.; Zhang, T.; Ma, Y.; Liu, J.; Yuan, B.; Chen, X.; Zhou, P.; Zhao, X.; Pang, F., et al. Mesenchymal stem cell sheets: a new cell-based strategy for bone repair and regeneration. Biotechnol Lett 2019, 41, 305-318, doi:10.1007/s10529-019-02649-7.

 [2] El-Ghannam, A. Bone reconstruction: from bioceramics to tissue engineering. Expert Review of Medical Devices 2005, 2, 87-101, doi:10.1586/17434440.2.1.87.

[3] Haugen, H.J.; Lyngstadaas, S.P.; Rossi, F.; Perale, G. Bone grafts: which is the ideal biomaterial? Journal of Clinical Periodontology 2019, 46, 92-102, doi:https://doi.org/10.1111/jcpe.13058.

These traditional BTE approaches provided successful results in clinics as they satisfied the macroscopic structural features and mechanical properties required for bone substitutes [4].

[4] none

[4] Gao, C.; Peng, S.; Feng, P.; Shuai, C. Bone biomaterials and interactions with stem cells. Bone Research 2017, 5, 17059, doi:10.1038/boneres.2017.59.

However, a lack of proper osteointegration during a long period was observed in the ma-jority of cases, requiring the repetition of surgical interventions in a lifetime [5].

[5] Albrektsson, T.; Johansson, C. Osteoinduction, osteoconduction and osseointegration. Eur Spine J 2001, 10 Suppl 2, S96-101, doi:10.1007/s005860100282.

[5] Gómez-Barrena, E.; Rosset, P.; Lozano, D.; Stanovici, J.; Ermthaller, C.; Gerbhard, F. Bone fracture healing: Cell therapy in delayed unions and nonunions. Bone 2015, 70, 93-101, doi:https://doi.org/10.1016/j.bone.2014.07.033.

Therefore, osteoconductive environments should include both mitogenic and angiogenic signaling molecules such as insulin-like growth factor (IGF I, II), fibroblast growth factor (FGF), transforming growth factor beta (TFG-b), and platelet-derived growth factor (PDGF) [2,11,12].

[2] Albrektsson, T.; Johansson, C. Osteoinduction, osteoconduction and osseointegration. Eur Spine J 2001, 10 Suppl 2, S96-101, doi:10.1007/s005860100282.

[11] None

[12] None

[2] El-Ghannam, A. Bone reconstruction: from bioceramics to tissue engineering. Expert Rev Med Devices 2005, 2, 87-101, doi:10.1586/17434440.2.1.87.

[11] Rao, R.R.; Stegemann, J.P. Cell-based approaches to the engineering of vascularized bone tissue. Cytotherapy 2013, 15, 1309-1322, doi:10.1016/j.jcyt.2013.06.005.

[12]              Devescovi, V.; Leonardi, E.; Ciapetti, G.; Cenni, E. Growth factors in bone repair. Chir Organi Mov 2008, 92, 161-168, doi:10.1007/s12306-008-0064-1.

At the initial part of the bone healing response, the osteoinduction process instantaneous-ly commences and dynamically directs complex but highly ordered signaling mecha-nisms, which induce 1) recruitment of MSCs and osteogenic cells, 2) differentiation of MSCs into bone-forming osteoblasts, and 3) stimulation of MSCs to secrete osteoinductive and osteoconductive factors to further support bone restoration (Figure 2B) [2,22].

[2] Albrektsson, T.; Johansson, C. Osteoinduction, osteoconduction and osseointegration. Eur Spine J 2001, 10 Suppl 2, S96-101, doi:10.1007/s005860100282.

[22] Marsell, R.; Einhorn, T.A. The biology of fracture healing. Injury 2011, 42, 551-555, doi:https://doi.org/10.1016/j.injury.2011.03.031.

[2] El-Ghannam, A. Bone reconstruction: from bioceramics to tissue engineering. Expert Rev Med Devices 2005, 2, 87-101, doi:10.1586/17434440.2.1.87.

[22] Marsell, R.; Einhorn, T.A. The biology of fracture healing. Injury 2011, 42, 551-555, doi:https://doi.org/10.1016/j.injury.2011.03.031.

With a growing interest in the therapeutic potential of MSCs, clinical research regarding MSC cell-based therapies have focused on the trophic activities of MSCs that can organize a regenerative microenvironment containing large amounts of bioactive molecules secret-ed by them [25].

[25] Caplan, A.I. All MSCs Are Pericytes? Cell Stem Cell 2008, 3, 229-230, doi:https://doi.org/10.1016/j.stem.2008.08.008.

[25] Oh, I.H. Mesenchymal stromal cells: new insight on their identity and potential role in cell therapy. Korean J Hematol 2010, 45, 219-221, doi:10.5045/kjh.2010.45.4.219.

As the implanted MSCs share angiogenic signaling crosstalk with ECs while expressing vascular endothelial growth factor (VEGF), FGF-2, angiopoietin 1 (Ang-1), and epidermal growth factors (EGFs), the resulting outgrowth of ECs is combined with the host vascular tissues[23,28].

[23] Dominici, M.; Le Blanc, K.; Mueller, I.; Slaper-Cortenbach, I.; Marini, F.; Krause, D.; Deans, R.; Keating, A.; Prockop, D.; Horwitz, E. Minimal criteria for defining multipotent mesenchymal stromal cells. The International Society for Cellular Therapy position statement. Cytotherapy 2006, 8, 315-317.

[28] Gallina, C.; Turinetto, V.; Giachino, C. A New Paradigm in Cardiac Regeneration: The Mesenchymal Stem Cell Secretome. Stem Cells International 2015, 2015, 765846, doi:10.1155/2015/765846.

[23] Martino, M.M.; Briquez, P.S.; Maruyama, K.; Hubbell, J.A. Extracellular matrix-inspired growth factor delivery systems for bone regeneration. Adv Drug Deliv Rev 2015, 94, 41-52, doi:10.1016/j.addr.2015.04.007.

[28] Almubarak, S.; Nethercott, H.; Freeberg, M.; Beaudon, C.; Jha, A.; Jackson, W.; Marcucio, R.; Miclau, T.; Healy, K.; Bahney, C. Tissue engineering strategies for promoting vascularized bone regeneration. Bone 2016, 83, 197-209, doi:10.1016/j.bone.2015.11.011.

MSCs secret multiple angiogenic factors including VEGF and increase the survival and growth of ECs, thus affecting rapid vascularization in many types of tissues [52].

[52] Takebe, T.; Enomura, M.; Yoshizawa, E.; Kimura, M.; Koike, H.; Ueno, Y.; Matsuzaki, T.; Yamazaki, T.; Toyohara, T.; Osafune, K., et al. Vascularized and Complex Organ Buds from Diverse Tissues via Mesenchymal Cell-Driven Condensation. Cell Stem Cell 2015, 16, 556-565, doi:10.1016/j.stem.2015.03.004.

[52] Beckermann, B.M.; Kallifatidis, G.; Groth, A.; Frommhold, D.; Apel, A.; Mattern, J.; Salnikov, A.V.; Moldenhauer, G.; Wagner, W.; Diehlmann, A., et al. VEGF expression by mesenchymal stem cells contributes to angiogenesis in pancreatic carcinoma. Br J Cancer 2008, 99, 622-631, doi:10.1038/sj.bjc.6604508.

As osteogenesis of MSCs progresses, the migration of EPCs is reduced due to the lack of EPC chemoattractants such as VEGF and FGF, which are actively secreted by undifferentiated MSCs [59].

[59] Li, R.; Nauth, A.; Gandhi, R.; Syed, K.; Schemitsch, E.H. BMP-2 mRNA expression after endothelial progenitor cell therapy for fracture healing. J Orthop Trauma 2014, 28 Suppl 1, S24-27, doi:10.1097/bot.0000000000000071.

[-] Yamamoto, K.; Ando, J. Emerging Role of Plasma Membranes in Vascular Endothelial Mechanosensing. Circ J 2018, 82, 2691-2698, doi:10.1253/circj.CJ-18-0052.

[59] Li, Z.; Yang, A.; Yin, X.; Dong, S.; Luo, F.; Dou, C.; Lan, X.; Xie, Z.; Hou, T.; Xu, J., et al. Mesenchymal stem cells promote endothelial progenitor cell migration, vascularization, and bone repair in tissue-engineered constructs via activating CXCR2-Src-PKL/Vav2-Rac1. The FASEB Journal 2018, 32, 2197-2211, doi:https://doi.org/10.1096/fj.201700895R.

In contrast, direct cell-cell contacts via coupling of gap junction proteins trigger more intimate cellular responses between MSCs and ECs [61].

[61] Yamamoto, K.; Ando, J. Emerging Role of Plasma Membranes in Vascular Endothelial Mechanosensing. Circ J 2018, 82, 2691-2698, doi:10.1253/circj.CJ-18-0052.

[61] Okamoto, T.; Suzuki, K. The Role of Gap Junction-Mediated Endothelial Cell-Cell Interaction in the Crosstalk between Inflammation and Blood Coagulation. International journal of molecular sciences 2017, 18, 2254, doi:10.3390/ijms18112254.

In addition, angiogenic potential of EPCs appeared to be reduced due to the lack of connexin 43, resuting in improper bone remodeling process [63].

[63] Okamoto, T.; Usuda, H.; Tanaka, T.; Wada, K.; Shimaoka, M. The Functional Implications of Endothelial Gap Junctions and Cellular Mcechanics in Vascular Angiogenesis. Cancers (Basel) 2019, 11, 237, doi:10.3390/cancers11020237.

[63] Wang, H.-H.; Su, C.-H.; Wu, Y.-J.; Li, J.-Y.; Tseng, Y.-M.; Lin, Y.-C.; Hsieh, C.-L.; Tsai, C.-H.; Yeh, H.-I. Reduction of connexin43 in human endothelial progenitor cells impairs the angiogenic potential. Angiogenesis 2013, 16, 553-560.

Scaffolds are 3D frameworks made of biomaterials that can induce and regulate cellular attachment, proliferation, migration, and differentiation (Figure 4) [29,64,65].

[29] none

[64] Rodrigues, M.T.; Gomes, M.E.; Reis, R.L. Current strategies for osteochondral regeneration: from stem cells to pre-clinical approaches. Curr Opin Biotechnol 2011, 22, 726-733, doi:10.1016/j.copbio.2011.04.006.

[65] Kolk, A.; Handschel, J.; Drescher, W.; Rothamel, D.; Kloss, F.; Blessmann, M.; Heiland, M.; Wolff, K.D.; Smeets, R. Current trends and future perspectives of bone substitute materials - from space holders to innovative biomaterials. J Craniomaxillofac Surg 2012, 40, 706-718, doi:10.1016/j.jcms.2012.01.002.

[29] García, J.R.; García, A.J. Biomaterial-mediated strategies targeting vascularization for bone repair. Drug Deliv Transl Res 2016, 6, 77-95, doi:10.1007/s13346-015-0236-0.

[64] Kolk, A.; Handschel, J.; Drescher, W.; Rothamel, D.; Kloss, F.; Blessmann, M.; Heiland, M.; Wolff, K.D.; Smeets, R. Current trends and future perspectives of bone substitute materials - from space holders to innovative biomaterials. J Craniomaxillofac Surg 2012, 40, 706-718, doi:10.1016/j.jcms.2012.01.002.

[65] Rodrigues, M.T.; Gomes, M.E.; Reis, R.L. Current strategies for osteochondral regeneration: from stem cells to pre-clinical approaches. Curr Opin Biotechnol 2011, 22, 726-733, doi:10.1016/j.copbio.2011.04.006.

Traditionally, various bioceramic materials such as hydroxyapatite, beta-tricalcium phosphate, and calcium phosphate cements have often been used for BTE because they exhibit structural and compositional similarities to mineralized bone tissues [3,67].

[3] El-Ghannam, A. Bone reconstruction: from bioceramics to tissue engineering. Expert Rev Med Devices 2005, 2, 87-101, doi:10.1586/17434440.2.1.87.

[67] Cordonnier, T.; Sohier, J.; Rosset, P.; Layrolle, P. Biomimetic materials for bone tissue engineering–state of the art and future trends. Advanced Engineering Materials 2011, 13, B135-B150.

[3] Haugen, H.J.; Lyngstadaas, S.P.; Rossi, F.; Perale, G. Bone grafts: which is the ideal biomaterial? Journal of Clinical Periodontology 2019, 46, 92-102, doi:https://doi.org/10.1111/jcpe.13058.

[67] Cordonnier, T.; Sohier, J.; Rosset, P.; Layrolle, P. Biomimetic materials for bone tissue engineering–state of the art and future trends. Advanced Engineering Materials 2011, 13, B135-B150.

Previous studies revealed that the fate of MSCs could be regulated by exposing them to a specific matrix stiffness that mimics a certain type of tissue in the absence of inducible growth factors [86-88].

[86] Wang, J.; Ye, Y.; Tian, H.; Yang, S.; Jin, X.; Tong, W.; Zhang, Y. In vitro osteogenesis of human adipose-derived stem cells by coculture with human umbilical vein endothelial cells. Biochem Biophys Res Commun 2011, 412, 143-149, doi:10.1016/j.bbrc.2011.07.062.

[87] Yourek, G.; McCormick, S.M.; Mao, J.J.; Reilly, G.C. Shear stress induces osteogenic differentiation of human mesenchymal stem cells. Regen Med 2010, 5, 713-724, doi:10.2217/rme.10.60.

[88] Gruene, M.; Pflaum, M.; Deiwick, A.; Koch, L.; Schlie, S.; Unger, C.; Wilhelmi, M.; Haverich, A.; Chichkov, B.N. Adipogenic differentiation of laser-printed 3D tissue grafts consisting of human adipose-derived stem cells. Biofabrication 2011, 3, 015005, doi:10.1088/1758-5082/3/1/015005.

[86] Yourek, G.; McCormick, S.M.; Mao, J.J.; Reilly, G.C. Shear stress induces osteogenic differentiation of human mesenchymal stem cells. Regen Med 2010, 5, 713-724, doi:10.2217/rme.10.60.

[87] Park, J.S.; Chu, J.S.; Tsou, A.D.; Diop, R.; Tang, Z.; Wang, A.; Li, S. The effect of matrix stiffness on the differentiation of mesenchymal stem cells in response to TGF-β. Biomaterials 2011, 32, 3921-3930, doi:10.1016/j.biomaterials.2011.02.019.

[88] Wingate, K.; Floren, M.; Tan, Y.; Tseng, P.O.; Tan, W. Synergism of matrix stiffness and vascular endothelial growth factor on mesenchymal stem cells for vascular endothelial regeneration. Tissue Eng Part A 2014, 20, 2503-2512, doi:10.1089/ten.TEA.2013.0249.

When cells are affected by mechanical stimulation, they can translate the strength to elec-trochemical signals, described as mechanotransduction [93,94].

[93] Hsieh, H.J.; Liu, C.A.; Huang, B.; Tseng, A.H.; Wang, D.L. Shear-induced endothelial mechanotransduction: the interplay between reactive oxygen species (ROS) and nitric oxide (NO) and the pathophysiological implications. J Biomed Sci 2014, 21, 3, doi:10.1186/1423-0127-21-3.

[94] none

[93] Hsieh, H.J.; Liu, C.A.; Huang, B.; Tseng, A.H.; Wang, D.L. Shear-induced endothelial mechanotransduction: the interplay between reactive oxygen species (ROS) and nitric oxide (NO) and the pathophysiological implications. J Biomed Sci 2014, 21, 3, doi:10.1186/1423-0127-21-3.

[94] Yamamoto, K.; Ando, J. Emerging Role of Plasma Membranes in Vascular Endothelial Mechanosensing. Circ J 2018, 82, 2691-2698, doi:10.1253/circj.CJ-18-0052.

Since the 1980s, when 3D-printing technologies were first introduced, manufacturing in-dustries have obtained considerable advantages to address the global demand for the customized fabrication of products with complex geometries and highly ordered archi-tectures [122].

[122] Zhang, B.; Pei, X.; Zhou, C.; Fan, Y.; Jiang, Q.; Ronca, A.; D'Amora, U.; Chen, Y.; Li, H.; Sun, Y., et al. The biomimetic design and 3D printing of customized mechanical properties porous Ti6Al4V scaffold for load-bearing bone reconstruction. Materials & Design 2018, 152, 30-39, doi:https://doi.org/10.1016/j.matdes.2018.04.065.

[122] Jovic, T.H.; Combellack, E.J.; Jessop, Z.M.; Whitaker, I.S. 3D Bioprinting and the Future of Surgery. Front Surg 2020, 7, 609836-609836, doi:10.3389/fsurg.2020.609836.

Round 2

Reviewer 2 Report

I would like to thanks the authors for the efforts made to modify their work. with this new insight, this work is more focus on bone healing and bring a lot of informations to help understanding of bone tissue engineering. 

the bibliography has been profoundly revised and is currently consistent with the authors' assertions. I therefore believe that this work deserves to be published.